# Immunothrombosis and the Role of Platelets in Venous Thromboembolic Diseases

**DOI:** 10.3390/ijms232113176

**Published:** 2022-10-29

**Authors:** Marco Heestermans, Géraldine Poenou, Anne-Claire Duchez, Hind Hamzeh-Cognasse, Laurent Bertoletti, Fabrice Cognasse

**Affiliations:** 1INSERM, U1059, SAINBIOSE, Jean Monnet University, F-42023 Saint-Etienne, France; 2French Blood Establishment (EFS) Auvergne Rhône Alpes-Scientific Department, F-42270 Saint-Etienne, France; 3Service de Médecine Vasculaire et Thérapeutique (Department of Vascular and Therapeutic Medicine), CHU de St-Etienne (St-Etienne University Hospital), F-42055 Saint-Etienne, France; 4INSERM, CIC-1408, CHU Saint-Etienne, F-42055 Saint-Etienne, France

**Keywords:** platelets, venous thrombosis, immunothrombosis, inflammation

## Abstract

Venous thromboembolism (VTE) is the third leading cardiovascular cause of death and is conventionally treated with anticoagulants that directly antagonize coagulation. However, recent data have demonstrated that also platelets play a crucial role in VTE pathophysiology. In the current review, we outline how platelets are involved during all stages of experimental venous thrombosis. Platelets mediate initiation of the disease by attaching to the vessel wall upon which they mediate leukocyte recruitment. This process is referred to as immunothrombosis, and within this novel concept inflammatory cells such as leukocytes and platelets directly drive the progression of VTE. In addition to their involvement in immunothrombosis, activated platelets can directly drive venous thrombosis by supporting coagulation and secreting procoagulant factors. Furthermore, fibrinolysis and vessel resolution are (partly) mediated by platelets. Finally, we summarize how conventional antiplatelet therapy can prevent experimental venous thrombosis and impacts (recurrent) VTE in humans.

## 1. Introduction

Platelets are small anucleated cells that are fragmented from megakaryocytes and released into the circulation. In healthy individuals, platelet levels range from 150,000 to 450,000 platelets per microliter of blood [1]. Many different functions are attributed to platelets and they are implicated in processes such as thrombosis and hemostasis, inflammation and regulation of tumor growth (reviewed in [2,3]). In the field of thrombosis and hemostasis, for many years platelets have been the main clinical target for managing arterial thrombotic diseases [4]. In contrast, platelets are believed to play a less prominent role during thrombosis within the venous system. Indeed, the vast platelet aggregates found in arterial thrombi are absent in venous thrombi [5]. Therefore, antiplatelet therapy is not commonly prescribed in the treatment of (recurrent) Venous Thrombo-Embolism (VTE), and anticoagulant drugs are more effective in combatting the disease. Anticoagulants inhibit secondary hemostasis or coagulation. The main driver of coagulation is thrombin, which cleaves the circulating glycoprotein fibrinogen to fibrin [6]. Fibrin can be crosslinked to generate large fibrin strands that can entangle blood cells leading to large thrombi that locally disrupt blood flow causing Deep Vein Thrombosis (DVT) [7]. When a fragment of the formed thrombus dislodges, it embolizes through the blood stream towards the lungs. This condition is known as Pulmonary Embolism (PE). VTE, the collective term for DVT and PE, affects 1–3 per 1000 individuals per year, and is the third leading cardiovascular cause of death [8]. 

The pathophysiology of venous thrombosis is different from that of arterial thrombosis, although an increasing overlap has been identified in recent years [9,10]. Venous thrombi are formed over a prolonged period (minutes versus hours/days). Initially, they mainly comprise erythrocytes and fibrin with the thrombi lacking the vast platelet aggregates found in larger quantities in arterial thrombi [11,12,13]. During the second stage of venous thrombosis, the fibrin gradually becomes dominant and replaces the blood cells. In the third stage of the disease, the thrombus is resolved in two phases. Collagen initially replaces the fibrinolyzing thrombus after which it is resolved in a process called collagenolysis. 

In recent years, platelets have been viewed as mediators of immunothrombosis, a novel concept that directly links thrombotic diseases including VTE to inflammatory processes [14]. It is a well-documented fact that platelets play a pivotal role in inflammation [15,16]. Firstly, platelets express numerous receptors and store hundreds of secretory products that are instrumental in functional responses. All of these components provide potential new routes for drug targeting to combat inflammatory disorders [17,18,19]. Secondly, platelets are widely recognized as secretors of proinflammatory cytokines, chemokines and biological response modifiers, such as the CD40 ligand [20]. Thirdly, platelets secrete lipid mediators that can act as autocrine and/or paracrine mediators, although the related intracellular signaling pathways are less clear than in nucleated cells and have yet to be clarified [21,22]. 

Starting out as a relatively innocent bystander, platelets have been converted into a potentially crucial driver of inflammation and appear to be critical for immunothrombosis. This review aims to provide an overview of the current literature in which platelets have been linked to in vivo venous thrombosis. The role of platelets during different aspects of venous thrombosis will then be outlined from initial inflammation of the vessel wall and the recruitment of leukocytes to platelet activators known to be directly involved in thrombus formation. Furthermore, the role of platelets in coagulation, fibrinolysis, thrombus resolution and vessel restauration will be covered. Finally, the efficacy of antiplatelets in (pre)-clinical VTE will be summarized. 

## 2. Platelets Interact with the Venous Vessel Wall

Venous thrombosis in large veins is initiated with the local inflammation of venous endothelial cells, which causes locally increased expression of surface selectins and VWF [23,24]. VWF is a multimeric glycoprotein produced by endothelial cells and stored in Weibel Palade bodies (WPBs) that reside in the cytoplasm [25]. Upon cellular stress, such as hypoxia or contact with inflammatory cytokines, WBPs fuse with the cellular membrane leading to the secretion of VWF. VWF functions in the circulation as a carrier protein for coagulation factor VIII (FVIII) by protecting the protease from rapid degradation, thus VWF and FVIII plasma levels are correlated [26]. FVIII is crucial for normal coagulation and a deficiency thereof results in hemophilia A, a severe bleeding disorder characterized by hemorrhages after vascular injury, deep bruising and joint pain and swelling [27]. Individuals with normal FVIII but a mutated form of VWF that lost its affinity for FVIII present a hemophilia A-like phenotype, emphasizing the critical role of VWF in stabilizing circulating FVIII. In addition to its role as a carrier protein, VWF mediates several other processes including primary hemostasis by directing platelet adhesion via its A1 domain that interacts with the platelet receptor glycoprotein GPIb [28] (Figure 1). Since the VWF A1 domain is exposed only upon high hemodynamic sheer stress, the consensus was that endothelial VWF-dependent platelet adhesion is not crucial for events in the venous vasculature. However, in mouse models for venous thrombosis based on the (partial) ligation of the inferior vena cava (IVC), VWF^−/−^ mice were protected from the disease [29]. Interestingly, when mice were supplemented with FVIII, thromboprotection was maintained, thereby implying that platelet-VWF interaction is pivotal for thrombus formation. In addition, infusion of an antibody blocking platelet interaction with the VWF A1 domain prevented thrombosis in wild-type mice. In the event of an injury within the venous vasculature, circulating VWF can also directly interact with the exposed subendothelial collagen via its A3 domain [30,31]. This interaction exposes the A1 domain that once again allows recruitment of circulating platelets via GPIb. 

Podoplanin is a well conserved, mucin-type transmembrane protein and is expressed in the vessel wall, although it is currently not entirely clear which precise cell type(s) express(es) the protein [32]. During hypoxia, a process comprising local stenosis and a risk factor for thrombus formation, the protein is upregulated. C-type lectin-like receptor 2 (CLEC-2) is a platelet receptor acting as a ligand for podoplanin. Indeed, platelet-specific CLEC-2^−/−^ mice were protected from venous thrombosis in a stenosis model of the IVC [33]. Conversely, blocking podoplanin using a specific antibody significantly decreased thrombus formation. Interestingly, a recent study showed that deletion of the S1P exporter Mfsd2b impeded venous thrombosis, possibly through decreased expression of podoplanin in the vessel wall of the IVC [34]. Podoplanin is a molecule that is also expressed by certain types of cancer cells. Cancers in which high levels of podoplanin are expressed are associated with an increased risk of developing cancer-associated thrombosis, the most common cause of cancer-related death after cancer itself [35,36]. Pre-clinical studies have shown that blocking podoplanin decreased thrombosis onset in cancer models, thus demonstrating that targeting podoplanin in cancer patients may be a feasible strategy in reducing thrombosis [37,38,39]. 

Damage of the venous vessel wall is one of the best-documented risk factors for VTE, and a specific trigger such as surgery or trauma can lead to thrombosis in large veins. When subendothelial cells expressing the highly prothrombotic coagulation factor and tissue factor (TF) are exposed in the circulation, coagulation is catalyzed via the extrinsic coagulation pathway. The platelet receptor, GPVI, displays strong affinity for subendothelial collagen, and in certain clinical scenarios this interaction may contribute to platelet recruitment in the veins. Antibody-inhibited or GPVI^−/−^ mice are less inclined to develop venous thrombosis [40,41]. However, it should be taken into account that platelet GPVI also binds to fibrin, which amplifies thrombin generation and maintains platelet recruitment [42]. Blocking platelet GPVI is linked to clot instability, possibly due to its interaction with fibrin [43,44]. 

## 3. Platelets Interact with and/or Recruit Other Immune Cells

The concept of immunothrombosis has been introduced in the last decade to describe the link between immunology and thrombotic processes in diseases such as viral and bacterial infections, and arterial thrombosis [45,46,47]. Immune cells are now also recognized as drivers of the pathology in VTE. It is often forgotten that platelets also function as immune cells and their traditional and novel roles in maintaining hemostasis and mediating inflammation, respectively, are not strictly separated during immunothrombosis [14]. In fact, in venous thrombosis, platelets most likely act more as mediators of leukocytes and less as hemostatic cells, in contrast to arterial thrombosis. Neutrophils and monocytes are two additional immune cell populations of interest and are known to play an important role in venous thrombus formation, with platelets often acting as mediators [48,49]. 

Neutrophils are granulocytes that are rapidly recruited towards sites of (sterile) inflammation. They also appear to be among the first leukocytes to arrive at the site where thrombus formation is initiated [50] (Figure 1). Locally, neutrophils are able to release so-called Neutrophil Extracellular Traps (NETs) that consist of DNA, histones and antimicrobial proteins. NETs are prothrombotic web-like structures in the extracellular space and dismantling NETs appears to be a tempting novel antithrombotic strategy for further investigation [51,52]. Several research groups have shown that the use of DNase-1, a compound that hydrolyses extracellular DNA, can prevent venous thrombosis in animal models [48,53,54]. Interestingly, a recently published study claims that DNase not only hydrolyses DNA but also Adenosine TriPhosphate (ATP) and Adenosine DiPhosphate (ADP) into adenosine [13]. ATP and ADP are potent agonists of both neutrophils and platelets while adenosine inhibits neutrophil function. Whether DNase prevents venous thrombosis via hydrolysis of DNA in NETs or the platelet agonists ATP/ADP, or a combination of both, has yet to be confirmed. 

Platelets and neutrophils are closely interlinked as mediators of venous thrombosis, and several direct interactions between the two cell types have been described. In this instance, platelets mostly act as mediators to recruit or activate neutrophils. In addition to its role in platelet recruitment to the endothelium, platelet receptor GPIb interacts with the leukocyte complement receptor, Mac-1 (also known as αMβ2). Blocking this specific interaction using a GPIb specific antibody resulted in reduced neutrophil recruitment, decreased NET formation and lowered thrombus formation [48]. Conversely, blocking or a genetic deficiency of Mac-1 attenuated thrombosis in mouse models of arterial thrombosis and microvascular thrombosis [55]. It remains to be seen whether antagonizing Mac-1, also directly prevents venous thrombosis. 

Solute Carrier Family 44 Member 2 (SLC44A2) is a choline transporter on which a single nucleotide polymorphism (SNP) has been correlated with an increased risk of VTE in humans [56,57]. Individuals carrying the SNP HNA3a have a 30% higher risk of developing VTE compared to those carrying the HNA3b variant. This finding was of considerable interest since it was the first gene not directly implicated in hemostasis to be identified in a genome wide association study (GWAS) on VTE. Mechanistically, platelet-expressed SLC44A2 promotes thrombin-mediated platelet activation by transporting choline to the mitochondria, thereby increasing mitochondrial oxygen consumption and ATP production [58]. SLC44A2 on neutrophils appears to have two binding partners that are relevant for thrombosis. Firstly, neutrophil SLC44A2 can directly interact with the A1 domain on endothelial VWF upon which NETosis takes place, thereby promoting thrombosis [59]. It was shown that the SNP found in the original GWAS studies determines the affinity of this interaction, possibly explaining the slightly increased risk of VTE in individuals with HNA3a compared to HNA3b [60]. Secondly, platelets can act as an intermediate between neutrophils and endothelial VWF. Platelets can bind to decrypted VWF via GPIb, thereby activating the αIIbβ3 receptor. Subsequently, activated αIIbβ3 is detected by neutrophil-expressed SLC44A2 under low sheer stress, again resulting in increased NETosis [61]. 

P-selectin (P-sel) is expressed by activated platelets and is probably the most commonly used marker to demonstrate platelet activation in vitro, making use of techniques such as flow cytometry or ELISA. P-sel expressed on the platelet membrane can interact with both neutrophils and monocytes via their P-selectin glycoprotein ligand-1 (PSGL-1) receptor, upon which firm adhesion is achieved through the binding of Mac-1 [62]. Consequently, monocytes are triggered to produce TF-positive microvesicles that promote coagulation and venous thrombosis [63,64]. Platelet P-sel also stimulates NET formation by interacting with neutrophils [65], and inhibiting P-sel function prevented experimental DVT [66]. Von Brühl et al., showed that P-sel deficient mice were protected from venous thrombosis, while the transfusion of wild type platelets did not correct the thrombotic phenotype [48]. These data imply that platelet P-sel per se is not crucial for leukocyte recruitment and that P-sel expressed by endothelial cells alone is sufficient to guide and activate neutrophils and monocytes at the site of immunothrombosis. 

Additional direct or indirect interactions between platelets and neutrophils have been described, which possibly contribute to NETosis and thrombus formation. S1P exporter Mfsd2b deficient platelets demonstrated reduced interaction between platelets and neutrophils compared to wild type platelets [34]. In addition to supporting platelet activation and experimental arterial thrombosis, platelet proline-rich tyrosine kinase Pyk2 regulated platelet-induced NETosis [67]. Pyk2 deficient mice were protected from venous thrombosis induced by IVC ligation. Platelet membrane-expressed CD40L interacts with neutrophil CD40 and soluble CD40L promoted platelet-neutrophil interaction, leading to increased NET formation [68,69]. Mice deficient in platelet amyloid precursor protein (APP) demonstrated increased platelet-neutrophil aggregates, NETosis and developed larger venous thrombi in a venous thrombosis mouse model [70]. The absence of APP might boost the inflammatory status of thrombotic mice, as evidenced by elevated plasma C-reactive protein levels. 

Platelets can physically interact with other circulating immune cells such as neutrophils and monocytes, and this process most probably contributes to the onset of (experimental) thrombosis in the venous vasculature. Another mode via which platelets can mediate various physiological processes is the secretion of active substances in the extracellular space [18]. Upon activation, platelets usually secrete various products stored in their lysosomes or α- or dense granules, thereby presenting different roles in inflammation, hemostasis and wound healing, for instance [71]. Platelet-secreted components also appear to be involved in venous thrombosis pathophysiology. SNAP23 deficient mice fail to release the contents of their platelet granules [72]. Interestingly, these mice are protected from both venous and arterial thrombosis, and although the exact mechanism for thromboprotection is currently unclear, these data emphasize the physiological relevance of granule secretion in thrombosis. P-sel can be expressed by platelets on the membrane and is used as an activation marker. However, P-sel can also be secreted from the platelet’s α-granules (soluble P-sel [73]). Soluble P-sel stemming from either platelets or endothelial cells can attract leukocytes to a site of (sterile) injury and may serve as a biomarker for VTE in humans [74,75,76]. Secreted platelet factor 4 (PF4 or CXCL4) may serve as a biomarker for VTE in humans [77,78,79], although its direct involvement in venous thrombosis pathophysiology is not exactly known. Myeloid-related protein-14 (MRP-14), a member of the alarmin or danger-associated molecular pattern molecules family, can be secreted by both platelets and neutrophils [80]. Platelet MRP-14 mediated NET formation and MRP-14 deficiency protected mice from venous thrombosis, which was corrected when mice were transfused with wild type platelets. 

## 4. Platelet Activators Promoting Venous Thrombosis

Upon activation, platelets undergo a phenotypical change associated with a significant change in expression and secretion pattern [81]. In the context of coagulation and venous thrombosis, several components that modulate platelet activation have been identified. Toll-like receptors (TLRs) are expressed by various immune cells [82,83,84]. These receptors recognize Damage-Associated Molecular Patterns (DAMPs) or Pathogen-Associated Molecular Patterns (PAMPs), resulting in cells that become activated, generate oxygen and nitrogen radicals, and/or produce cytokines (Figure 1) [85]. Platelets also express TLRs at both their membrane and intracellular, and platelets are activated in response to DAMPs or PAMPs after which they mediate an immune response [82,86]. In mouse venous thrombosis, HMGB1 can function as a DAMP and is both secreted and recognized by platelets, presumably via TLR2, TLR4, and/or RAGE [87]. HMGB1 deficient mice or wild type mice treated with an HMGB1 inhibitor presented decreased thrombus formation, possibly because of reduced monocyte recruitment and signs of NETosis in the thrombus. In addition, supplementing mice with recombinant HMGB1 enhanced thrombus formation while platelet-specific HMGB1^−/−^ mice were protected from thrombosis, coinciding with a decrease in the number of neutrophils present in the formed thrombi [53]. Kindlin-3, an integrin activator expressed in platelets, and paxillin appeared crucial for αIIbβ3 inside-out signaling [88,89,90]. In mice, disruption of the interaction between kindlin-3 and paxillin or a selected deficiency for platelet-Kindlin-3 demonstrated a significant decrease in experimental DVT.

Thrombin is the major coagulation mediator, and is involved in both inflammatory and thrombotic processes [91]. In humans, thrombin serves as a potent platelet activator mainly via protease-activated receptors (PAR) 1 [92], and the PAR1-specific thrombin derivative TRAP (thrombin receptor-activating peptide) is often used to activate platelets in vitro. In mice, PAR4 is the main thrombin receptor and platelet-specific PAR4^−/−^ mice are indeed protected from venous thrombosis [93]. The complement cascade is an innate immune response that relies on the sequential activation of serine proteases, similar to the coagulation cascade. Platelets from complement factor C3 deficient mice displayed reduced platelet activation in response to a PAR4-specific stimulus [94]. In line with these results, Subramaniam et al., demonstrated that, upon activation, platelets from C3^−/−^ mice showed reduced VWF binding, P-sel exposure, αIIbβ3 activation and phosphatidyl serine (PS) exposure [95]. Additionally, in an IVC ligation model, C3^−/−^ mice were protected from thrombosis. Pro-inflammatory interleukins 9 and 17A promoted venous thrombosis in a mouse model based on IVC stasis, most likely due to direct stimulation of platelet activation [96,97]. Mice treated with estradiol demonstrated diminished platelet responsiveness and were protected from thromboembolic events and venous thrombosis [98,99]. Interestingly, estradiol treatment did not modulate the potency of the coagulation cascade, implying that the effects on thrombosis were platelet-dependent. Growth arrest-specific gene 6 (Gas6) is a vitamin-K dependent growth factor that is expressed and secreted by many cell types, including platelets [100]. Platelets from Gas6 deficient mice or mice with a platelet receptor deficiency (Tyro3, Axl and Mer) demonstrated a substantial decrease in the stabilization of platelet aggregates due to a disturbed inside-out signaling of αIIbβ3, resulting in protection from experimental venous thrombosis [101,102]. NADPH oxidase-derived reactive oxygen species (ROS) mediate normal platelet function and activation, and multiple studies have shown that altered ROS production modulates the risk for venous thrombosis, both in humans and in vivo models (reviewed in [103]). 

## 5. Platelets and the Coagulation Cascade

Platelets are involved in the initiation of venous thrombosis by directing leukocytes to the site of (sterile) inflammation. However, they also directly participate in coagulation or secondary haemostasis [104,105]. Coagulation is the process that shapes blood from a liquid to a gel-like substance, following a process usually visualized as a complex cascade of bioactive serine proteases resulting in fibrin formation. A blood clot is characterized by these polymerized fibrin strands that entangle other circulating cells. Activated platelets, which play a supporting role in coagulation, express increased levels of phosphatidylserine (PS) on their surface that can be used as a marker of platelet activation (Figure 2). Pro- and anticoagulant factors with a domain rich in gamma-carboxylated amino acids (the so-called GLA domain of vitamin K-dependent coagulation factors) are recruited to PS-rich surfaces where they are efficiently activated [106,107,108]. Platelet membrane coagulation affects the tenase complex (factor Xa and IXa, and co-factor VIIIa) and the prothrombinase complex (factor Xa and cofactor Va). In addition, anticoagulants proteins C and S exert their effects on the PS-positive platelet surface, where they inactivate cofactors Va and VIIIa.

Platelet Pyk2 regulates PS expression, possibly explaining the thromboprotection observed in Pyk2 deficient mice [67]. In a mouse model relying on the transient inhibition of two important natural anticoagulants, antithrombin and protein C, complete platelet depletion corrected the onset of severe thrombotic coagulopathy [109]. Since traces of subclinical fibrin formation were found in the liver of the platelet-depleted mice, it was hypothesized that the absence of platelets did not prevent thrombosis entirely but rather limited the progression of massive coagulation for which a PS surface is required. These data suggest that restricting platelet PS exposure may be one way of limiting venous thrombosis progression. 

In addition to the well-documented role of platelets in exposing PS to mediate coagulation, platelets also secrete several components that directly influence coagulation. Activated platelets are the main source of calcium required for the functioning of serine proteases such as thrombin and factor X [110]. In addition, recent studies have shown that platelets secrete several pro- and anticoagulants, such as fibrinogen [111], factor V [112,113], factor VIII [114,115], factor IX [116], factor XIII [117,118], VWF [119] and protein S [120]. All of these coagulation factors are not produced exclusively by platelets, hence the significance of the platelet-derived pool of each coagulation factor may vary. 

Upon activation, platelets shed 0.1–1-μm fragments that express functional receptors and are PS-positive. These cellular fragments are commonly referred to as microvesicles or microparticles, and it has been shown that platelet-derived microvesicles are omnipresent within the circulation [121,122]. PS-positive microvesicles can catalyze coagulation, possibly in locations where platelets cannot penetrate [123,124], and it has been shown that they are enriched around the developing thrombus site [125]. Platelet-derived microvesicles promote venous thrombosis in mice [126] and may be useful as biomarkers for VTE in humans [127]. Another challenge is to investigate the role of microvesicles in transporting micro RNAs (miRNA) in immunothrombotic diseases. miRNAs are non-coding and modulate gene expression through mRNA degradation or translational repression of numerous targets [128]. Understanding the heterogeneity of (platelet) microvesicles will help them become biomarkers and may reveal specific roles for each microparticle subtype in health and disease [129].

Polyphosphates (polyp) are another prothrombotic component secreted by platelets from their dense granules that mediate coagulation. Interestingly, it has been shown that polyp are both procoagulant and proinflammatory and may thus mediate immunothrombosis [130]. Polyp’s function and activity depends on its polymer chain length [131]. Long chain polyp is insoluble and, following platelet activation, the polymer is retained on the surface in nanoparticles with divalent metal ions [132]. Here, it locally supports FXII activation leading to bradykinin formation and thrombin formation following the intrinsic pathway. A polyp-neutralizing agent prevents FXII-dependent mouse venous thrombosis, emphasizing its physiological relevance and clinical potential in preventing or curing VTE in humans [133]. Platelet-secreted soluble short chain polyp does not efficiently activate FXII. However, this form is able to modulate coagulation by supporting thrombin-dependent FXI activation [134] and FV activation [135], and also modifies fibrin polymerization [136].

## 6. Platelets during Thrombus Resolution and Vessel Restoration

Thrombus resolution starts with the process of breaking down crosslinked fibrin strands in a process called fibrinolysis. Defective fibrinolysis can lead to several chronic and acute pathologies, mostly linked to haematological disorders [137]. The main fibrinolysin is plasmin, a protease resulting from the cleavage of inactive circulating plasminogen. Comparable to coagulation, fibrinolysis is tightly controlled with several specific proteases, cofactors and protease inhibitors that all have a part to play in ensuring correct thrombus fibrinolysis. Platelets provide the PS-positive surface to allow efficient fibrinolysis (Figure 2). In addition, several reports show that platelets store and secrete both fibrinolytics, such as plasminogen [138,139] and antifibrinolytic agents including the plasminogen activator inhibitor-1 [140] and thrombin activatable fibrinolysis inhibitor [141]. Secretion of these factors make platelets key mediators of fibrinolysis.

Thrombus maturation, resolution and post-thrombotic vessel wall remodeling are the final stages in the lifespan of a thrombus, and platelets are also involved [142]. In a murine model for venous thrombosis, platelet depletion was seen to decrease thrombus fibrosis, smooth muscle cell invasion and intimal vessel wall thickening-a recognized histological feature associated with post-thrombotic syndrome. It was suggested that platelets stimulate smooth muscle cell invasion by secreting TGF-β, bFGF and PDGF, all of which are growth factors associated with this process. Interestingly, mice experimentally infected with *Staphylococcus aureus* also demonstrated a misguided thrombus resolution, possibly mediated by upregulation of similar growth factors [143]. Mice treated with statin, a compound associated with decreased platelet activity, showed improved thrombus resolution [144]. Whether the antiplatelet effect is responsible for altering thrombus resolution has yet to be investigated, since statins are known to have pleiotropic effects including dampening immune responses [145]. The platelet endothelial cell adhesion molecule-1 (PECAM-1) is a protein expressed by many different cells including platelets [146]. PECAM-1 deficient mice also displayed increased platelet activation and misguided thrombus resolution, although the precise role of platelet-expressed PECAM-1 was not addressed in the case of the latter [147]. 

## 7. Platelet Inhibition to Prevent Venous Thrombosis

Platelets appear to play a pivotal role in the initiation, progression and resolution of experimental venous thrombosis. Indeed, it has been demonstrated that platelet inhibitors/antiplatelet drugs prevent venous thrombosis in different animal models. Currently prescribed as antiplatelet drugs are Acetylsalicylic acid (ASA, better known as aspirin), the P2Y12 receptor antagonists clopidogrel, prasugrel, and ticagrelor, and the αIIbβ3 receptor antagonist, abciximab [148]. Aspirin is an irreversible inhibitor of platelet aggregation, which prevents platelet thromboxane A2 synthesis by direct binding to COX-1 and COX-2 [149]. Although the exact mechanism of action has not been fully elucidated in recent decades, the drug has been used for thousands of years as an effective pain killer or anti-inflammatory drug since it blocks the production of prostaglandins that can serve as molecular pain signals. P2Y12 inhibitors are part of a second class of antiplatelet drugs [150]. These compounds also inhibit platelet aggregation by preventing interaction between ADP and the P2Y12 platelet receptor that normally results in platelet activation. Finally, the αIIbβ3 receptor antagonist, abciximab, reduces fibrinogen-mediated platelet-platelet interactions thereby preventing the formation of aggregates [151,152].

In a laser injury model in rats, aspirin treatment decreased the number of thrombi and emboli formed [153]. More recently, a mouse model based on IVC ligation showed that aspirin decreased thrombus size, presumably because of reduced monocyte and neutrophil activation leading to lower TF activity and NET formation [154]. Multiple studies have assessed the effects of P2Y12 antagonists, mainly clopidogrel, on experimental venous thrombosis. Clopidogrel appeared to be effective in preventing venous thrombosis in dogs, rabbits and rats [155,156,157,158]. In mice treated with ticagrelor, venous thrombosis as a result of complete IVC ligation was significantly decreased [159]. Thrombus weight as a result of a ferric-chloride induced vascular injury of the vena cava was lowered in P2Y12^−/−^ or wild type clopidogrel-treated mice compared to (untreated) wild type mice [160]. The accumulation of platelets in large vessel venous thrombosis in mice was decreased upon treatment with clopidogrel, while fibrin formation was not significantly altered [161]. In the mouse IVC stenosis model, aspirin and clopidogrel combined, but not alone, reduced thrombus formation [162]. Finally, the small molecule UNC2025, a compound that interferes with the Gas6 signalling pathway, decreased platelet activation alone or in combination with clopidogrel, and increased survival in a lethal PE mouse model [163]. 

In contrast to studies on experimental venous thrombosis, the efficacy of antiplatelet therapy in preventing human (secondary) VTE is still debated, and only aspirin has been tested in dedicated clinical studies. Current clinical practice guidelines from the American Academy of Orthopedic Surgeons, the American College of Chest Physicians, the American Society of Hematology and the National Institute of Health recommend aspirin as a potential thromboprophylactic agent, like Low Molecular Weight Heparin (LMWH) or Direct Oral AntiCoagulants (DOAC) without any preference expressed in terms of therapeutic options. In the EPCAT study, aspirin proved effective in preventing VTE after 5 days of treatment with the FXa inhibitor, rivaroxaban, in orthopedic surgery at the cost of a slightly increased risk of bleeding [164]. In contrast, the CRISTAL trial failed to demonstrate noninferiority of aspirin as compared to LMWH [165]. The results of another ongoing trial, the PEPPER trial (NCT02810704) compares aspirin to rivaroxaban and warfarin, and this trial may provide more inside on the use of aspirin to prevent primary VTE.

Two randomized clinical trials published in 2012 have highlighted encouraging results for aspirin in the secondary prevention or extended treatment of VTE. In the WARFASA and ASPIRE studies, the incidence of recurrent VTE was reduced by 40% and 35%, respectively [166,167]. In both studies, reduction of recurrent VTE coincided with a relatively low incidence of bleeding events. In 2017, Weitz et al., published the results of a randomized controlled phase 3 trial comparing rivaroxaban to aspirin for the extended treatment of VTE [168]. The incidence of adverse events was similar for both drugs but the risk of recurrence was lower with rivaroxaban. Consequently, aspirin was not validated for secondary prevention of VTE. In 2019, Mai et al., performed a meta-analysis of 18 independent randomized controlled trials, comparing the risk of recurrent thrombosis and major bleeding when treated with several antithrombotic drugs (LMWH, DOAC, VKA, or aspirin) [169]. For aspirin, a recurrence rate of 0.71 (0.55–0.91) was reported, which is about three times greater than the risk to patients treated with anticoagulants, there being no significant reduction in major bleeding. These data imply that the benefit of curing patients with recurrent thrombosis using aspirin is limited. 

In a study on patients treated with both rivaroxaban/dabigatran and antiplatelet drugs, it was shown that the improved efficacy to prevent atherothrombosis did not stem from a direct antiplatelet effect of DOACs but from their inhibitory effect on platelet aggregation secondary to coagulation activation. This effect may differ per DOAC, depending on the targeted coagulation factor [170]. These results were confirmed in a dedicated pharmacokinetic study on blood samples from patients receiving rivaroxaban, a DOAC targeting FXa, in which the effect of several platelets agonists (ADP, arachidonic acid, epinephrine, collagen and thrombin) at different doses administrated were tested [171]. Thrombin antagonist dabigatran may inhibit platelet activation, in particular by thrombin [172]. Finally, FXa inhibitors apixaban and edoxaban might have the capacity to indirectly inhibit platelet aggregation [173,174].

## 8. Conclusions

For decades, platelets have been regarded as crucial players in arterial thrombosis, and this disease can be efficiently prevented or cured by antiplatelet therapy. More recently, pre-clinical data have provided an overwhelming amount of evidence to suggest that platelets also play a pivotal role in venous thrombosis as mediators of immunothrombosis. Platelets are involved in initial vascular inflammation, platelet secretion of biological response modifiers, recruitment and/or activation of leukocytes, thrombus progression and resolution as well as vessel wall remodeling. In this review, we aimed to provide an overview of the known roles of platelets in experimental venous thrombosis, and to outline the implications in the human VTE context. The versatility of platelets, particularly in terms of their inflammatory properties, is a constant source of amazement and many questions remain unanswered regarding their role in experimental venous thrombosis, and how to extrapolate these in vivo results to the human pathology. Finally, depending on the clinical scenario, it will be very interesting to establish whether antiplatelet therapy is a viable option in addition to or as a replacement for anticoagulation therapy in VTE patients. 

## Figures and Tables

**Figure 1 ijms-23-13176-f001:**
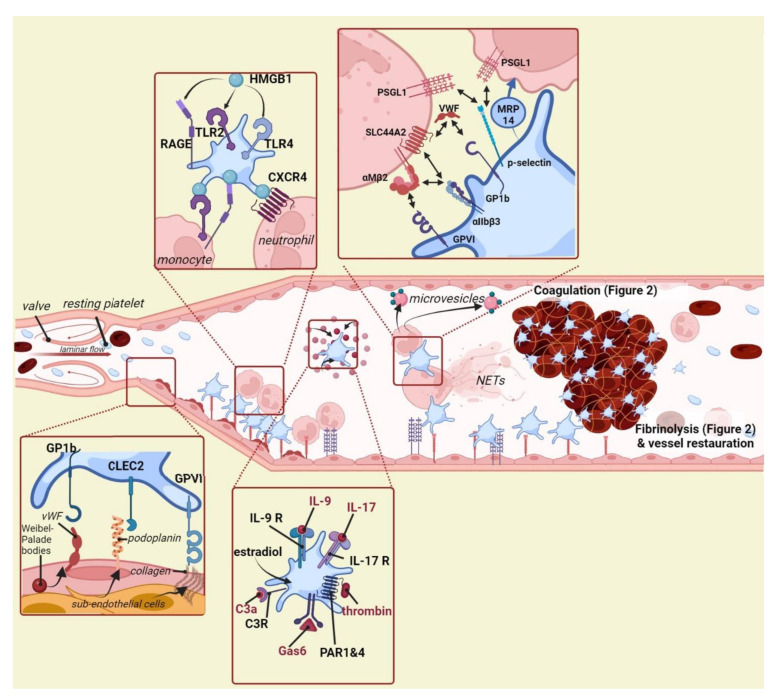
The role of platelets in different aspects of venous thrombosis. Platelets are crucially involved in the pathophysiology of venous thrombosis. They can be recruited to the vessel wall and can interact with neutrophils and monocytes. In addition, activated platelets can mediate venous thrombosis including via stimulation of coagulation and fibrinolysis (see Figure 2).NETs: neutrophil extracellular traps, vWF: von Willebrand factor, GP1b: Glycoprotein 1b, CLEC2: C-type lectin-like receptor 2, GPVI: Glycoprotein VI, RAGE: TLR: Toll-like receptor, HMGB1: High mobility group box 1, CXCR4: C-X-C chemokine receptor type 4, PSGL-1: P-selectin glycoprotein ligand-1, SLC44A2: Solute Carrier Family 44 Member 2, αMβ2: integrin αMβ2 or Mac-1 (macrophage 1 antigen), MRP14: migration inhibitory factor-related protein 14, IL: Interleukin, PAR1-4: Protease-activated receptor 1–4, C3a: Complement factor 3a, GAS6: Growth arrest–specific 6. Image is created using Biorender.com.

**Figure 2 ijms-23-13176-f002:**
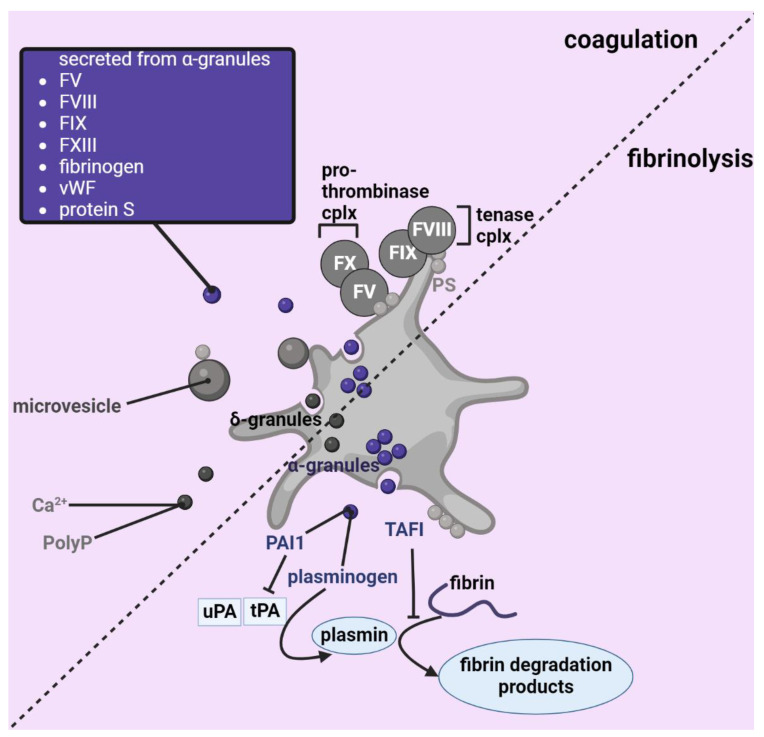
The role of platelets in coagulation and fibrinolysis. Platelets are involved in both coagulation and fibrinolysis. Upon activation, they can secrete microvesicles, coagulation and fibrinolysis factors from their α-granules, and Ca^2+^ and polyphosphates (polyp) from their dense granules. In addition, they expose phosphatidylserine on their membrane to support both coagulation and fibrinolysis. FV: coagulation factor V, FVIII: coagulation factor VIII, FIX: coagulation factor IX, FXIII: coagulation factor XIII, vWF: von Willebrand factor, FX: coagulation factor X, PAI1: plasminogen activator inhibitor 1, TAFI: thrombin activatable fibrinolysis inhibitor, uPA/tPA: urokinase/tissue plasminogen activator, δ-granules: dense granules. Image is created using Biorender.com.

## Data Availability

Not applicable.

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
