# Peer review of "Immunothrombosis and the Role of Platelets in Venous Thromboembolic Diseases"

_ijms, 2022, doi:10.3390/ijms232113176_

Round 1

Reviewer 1 Report

I have read with interest the review paper written by Heestermans et al. The authors review the role of platelets in Venous Thromboembolic Diseases (VTE). 

The paper is well-written but not well-organized. Therefore, I found it confusing. 

Here there are my suggestions for improvement:

The abstract does not communicate to the reader what the authors aim to review in the paper so I suggest clarifying it. 

The Introduction is unnecessary long. The second and third paragraphs are not required. I would only focus on VTE and not spend one paragraph on arterial thrombotic disease. Section 8 seems sufficient to discuss anti-platelet therapy. 

Section 2 seems out of place. I would insert it later in the paper (maybe between sections 7 and 8).

Figure 1 is well-done. However, it would be good to keep cell size comparable.  

Conclusion: in this section, I would discuss more thoroughly the challenges in studying platelets in VTE and add suggestions (if any) on how to improve our knowledge. The aim of the review, as suggested, would be more appropriate in the abstract.       

Author Response

I have read with interest the review paper written by Heestermans et al. The authors review the role of platelets in Venous Thromboembolic Diseases (VTE).

The paper is well-written but not well-organized. Therefore, I found it confusing. 

Here there are my suggestions for improvement:

The abstract does not communicate to the reader what the authors aim to review in the paper so I suggest clarifying it.

We have rewritten the abstract so it aligns better with the aims of our manuscript.

The Introduction is unnecessary long. The second and third paragraphs are not required. I would only focus on VTE and not spend one paragraph on arterial thrombotic disease. Section 8 seems sufficient to discuss anti-platelet therapy.

According to the reviewer’s suggestion, we have significantly shortened the introduction of the manuscript by discarding most of the information about arterial thrombotic diseases.

Section 2 seems out of place. I would insert it later in the paper (maybe between sections 7 and 8).

We decided to delete section 2, which discussed the role of animals models used to study the role of platelets in venous thrombosis. Some elements of the section are discussed throughout the manuscript and after a critical revision we deemed that other elements were not crucial for the manuscript and as the reviewer indicates may be confusing.  

Figure 1 is well-done. However, it would be good to keep cell size comparable. 

We adapted figure 1. First, within the “vessel” the cells now have a comparable size. Second, within two insets (top left and bottom left) we adjusted the layout of the cells other than platelets (monocyte, neutrophil, and endothelial cell) that they do not appear smaller or with a similar size as the platelet.

Conclusion: in this section, I would discuss more thoroughly the challenges in studying platelets in VTE and add suggestions (if any) on how to improve our knowledge. The aim of the review, as suggested, would be more appropriate in the abstract. 

The authors would like to thank the reviewer for the constructive comments, and we hope that the structural changes have improved the readability and comprehensibility of the manuscript.

Reviewer 2 Report

Correct, thorough summary of the role of platelets in venous thromboembolism. I have only a minor suggestion:

To 8. Platelet inhibition to prevent venous thrombosis - Considering that there is a significant difference in indication and application between DOACs, it would be useful for the reader to explain separately the potential interaction between individual DOACs (dabigatran, edoxaban, apixaban, etc.) and platelets in case of venous thrombolembolism, if there is literature on this data.

Author Response

Correct, thorough summary of the role of platelets in venous thromboembolism. I have only a minor suggestion:

To 8. Platelet inhibition to prevent venous thrombosis - Considering that there is a significant difference in indication and application between DOACs, it would be useful for the reader to explain separately the potential interaction between individual DOACs (dabigatran, edoxaban, apixaban, etc.) and platelets in case of venous thrombolembolism, if there is literature on this data.

The authors would like to thank the reviewer for the positive review of our manuscript. As suggested by the reviewer, we added a section in which potential interaction between DOACs and platelets are discussed (end of the section named “Platelet inhibition to prevent venous thrombosis”).

Round 2

Reviewer 1 Report

The authors addressed all my concerns! Good luck with your future work!